# Model-Decoupling-Based Federated Learning with Consistency via Knowledge Distillation Using Conditional Generator

## Abstract

Federated Learning (FL) is gaining popularity as a distributed learning framework that only shares model parameters or gradient updates and keeps private data locally. However, FL is at risk of privacy leakage caused by privacy inference attacks. And most existing privacy-preserving mechanisms in FL conflict with achieving high performance and efficiency. Therefore, we propose FedMD-CG, a novel FL method with highly competitive performance and high-level privacy preservation, which decouples each client's local model into a feature extractor and a classifier, and utilizes a conditional generator instead of the feature extractor to perform server-side model aggregation. To ensure the consistency of local generators and classifiers, FedMD-CG leverages knowledge distillation to train local models and generators at both the latent feature level and the logit level. Also, we construct additional classification losses and design new diversity losses to enhance client-side training. FedMD-CG is robust to data heterogeneity and does not require training extra discriminators (like cGAN). We conduct extensive experiments on various image classification tasks to validate the superiority of FedMD-CG. We provide our code here: https://anonymous.4open.science/r/FedMD-CG-34E2/.

## 1 Introduction

Many modern real-world applications involve data being dispersed across clients located in different physical locations, such as autonomous driving (Li et al., 2021), medical image analysis (Liu et al., 2021), and IoT (Nguyen et al., 2021). However, various regulation, privacy and security concerns often make it impractical or even impossible to collect these dispersed data into one location for traditional centralized learning (Voigt & Von dem Bussche, 2017). To ameliorate these limitations, Federated Learning (FL) (Li et al., 2020a) has been proposed to enable each client to train a local model using only its own data and share its model parameters or gradient updates with a central server periodically to ensure that each client's raw data does not leak from the local device.

Despite the success, the vanilla FL (e.g., FedAvg (McMahan et al., 2017) and its variants (Li et al., 2020b; Karimireddy et al., 2020; Luo et al., 2023)) based on sharing complete local model parameters or gradient updates are extremely vulnerable to inference attacks. Several prior arts empirically demonstrate that it is feasible to reconstruct victim clients' private data from trained and publicly shared parameters and gradient updates (Zhu et al., 2019; Geiping et al., 2020; Haim et al., 2022). Therefore, a variety of efforts have been devoted to reducing the risk of privacy leakage in FL, including homomorphic encryption (HE) (Ma et al., 2022; Zhang et al., 2022b), differential privacy (DP) (Geyer et al., 2017; Cheng et al., 2022) and model decoupling (MD) (Arivazhagan et al., 2019; Liang et al., 2020). In particular, HE achieves high-level privacy protection at the expense of extremely high computation and communication costs, which restricts its deployment in bandwidth-limited and large-model scenarios. DP preserves privacy by perturbing server-side model aggregation or client-side local model update, but this deteriorates the performance of the FL methods. See Appendix A for more related work.

In this paper, we mainly focus on MD, which requires each client to decompose the local model into the base and top layers, and send one of them to the server to reduce the risk of privacy leakage, yet this inevitably results in performance degradation and even privacy exposure. Note that we regard

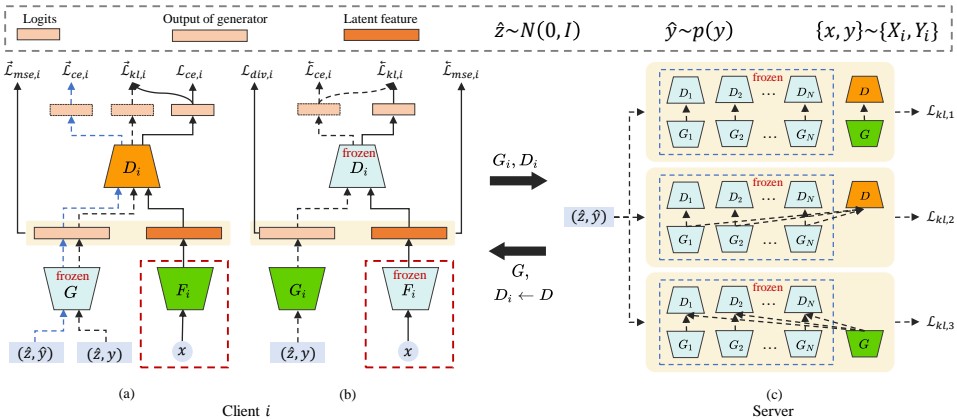

Figure 1: Illustration of FedMD-CG: (a) The local model update distills the experience from the global generator $G$ for augmenting the generalization performance of the local model $[F_i, D_i]$. (b) The local generator update utilizes the trained local model $[F_i, D_i]$ to guide the local generator $G_i$ to mimic latent feature space. Note that $G$ is not involved in client-side training. (c) The server-side data-free KD aggregation takes a crossed manner to achieve as much knowledge transfer as possible. Best viewed in color. Zoom in for details.

the base layers (top layers) as a feature extractor (classifier). Recently, FedCG (Wu et al., 2021) combines FL and conditional generative adversarial network (cGAN) (Mirza & Osindero, 2014) to adversarially train a conditional generator to replace the feature extractor for each client, aiming at achieving competitive performance while maintaining high-level privacy protection. However, we revisit it and observe that the following pitfalls may occur in client-side training. First, knowledge transfer modality at the latent feature level may not be sufficient. Second, additional discriminators need to be trained to satisfy the adversarial training of cGAN. Third, the trained local generator may not match the local classifier, terming their inconsistency. Note that the latent feature denotes the output of the feature extractor.

To this end, we propose a new **Fed**erated Learning with **MD** method (dubbed as FedMD-CG), which resorts to knowledge distillation (KD) to train a local **c**onditional **g**enerator for each client to replace the local feature extractor. To be more specific, FedMD-CG works on how to efficiently train the local model and generator on the client side. To achieve this, FedMD-CG utilizes KD to perform knowledge transfer from the global generator to the local model and from the local model to the local generator at the latent feature level and the logit level. Meanwhile, we additionally construct two classification losses to enhance the local model update and the local generator update, respectively. In addition, we devise two novel diversity constraints to ensure the diversity of the local generator outputs. On the server side, FedMD-CG performs aggregation of local generators and classifiers in a crossed data-free KD fashion. The overview of our method is illustrated in Fig. 1.

In a nutshell, the main contributions of this work are as follows: 1) We formulate a novel privacy-preserving FL method FedMD-CG to achieve better generalization performance, via leveraging KD to efficiently transfer knowledge from the global generator to the local model and then from the local model to the local generator. 2) To enhance client-side training, we construct additional classification losses and tailor new diversity constraints. Our method ensures the consistency between trained local generators and classifiers, thereby being robust to data heterogeneity. 3) FedMD-CG performs aggregation in a crossed data-free KD fashion on the server side in order to extract as much knowledge as possible from the local generators and classifiers. 4) We conduct extensive experiments to show that FedMD-CG is highly competitive compared with state-of-the-art baselines w.r.t test performance, convergence speed and privacy protection.

## 2 PROPOSED METHOD

In this section, we detail the proposed method FedMD-CG. We first define the problem setup and notations for clarity. And then the core modules of FedMD-CG are presented. Moreover, we present pseudocode for FedMD-CG in Appendix B.

**Problem Setup and Notations.** In this work, we consider supervised federated learning (FL) setting, i.e., the general problem of multi-class classification. To be specific, we focus on the centralized setup that consists of a central server and $N$ clients owning private labeled datasets $\{(\boldsymbol{X}_i, \boldsymbol{Y}_i)\}_{i=1}^N$ with $|\boldsymbol{X}_i| = n_i$, where $\boldsymbol{X}_i = \{\boldsymbol{x}_i^b\}_{b=1}^{n_i}$ follows data distribution $\mathcal{D}_i$ over input feature space $\mathcal{X}_i$, i.e., $\boldsymbol{x}_i^b \sim \mathcal{D}_i$, and $\boldsymbol{Y}_i = \{y_i^b\}_{b=1}^{n_i} \subset \mathbb{R}$ denotes the ground-truth labels of $\boldsymbol{X}_i$. Notably, we consider the same input feature space, yet the sample distribution may be different among clients, that is, data heterogeneity caused by label distribution skewness (i.e., $\mathcal{X}_i = \mathcal{X}_j$ and $\mathcal{D}_i \neq \mathcal{D}_j, \forall i \neq j, i, j \in [N]$). Besides, each client $i$ holds a local model parameterized by $\boldsymbol{\theta}^i = [\boldsymbol{\theta}_F^i; \boldsymbol{\theta}_D^i]$ comprising two components: the base layers (feature extractor) $F_i : \boldsymbol{X}_i \to \mathcal{F}$ parameterized by $\boldsymbol{\theta}_F^i$, and the top layers (classifier) $D_i : \mathcal{F} \to \mathbb{R}^c$ parameterized by $\boldsymbol{\theta}_D^i$, where $\mathcal{F} \subset \mathbb{R}^p$ is the output space of feature extractor with $p$ dimension, i.e., the latent feature space, and $c$ is the number of classes. For extracting knowledge from clients without accessing any extra data, each client equips with a conditional generator $G : \mathcal{Z} \times \mathcal{Y} \to \mathbb{R}^p$ parameterized by $\boldsymbol{w}$, where $\mathcal{Z} \subset \mathbb{R}^q$ is the multivariate standard normal distribution $\mathcal{N}(\boldsymbol{0}, \boldsymbol{I})$ and $\mathcal{Y} \subset \mathbb{R}^c$ indicates the one-hot vector space of the ground-truth label. We use bold $\boldsymbol{y} \in \mathcal{Y}$ to denote one-hot vector corresponding to class $y \in \mathbb{R}$. Hereafter, we refer to conditional generator as generator.

## 2.1 CLIENT-SIDE TWO-STAGE DISTILLATION

The training process for each client $i$ involves two stages: augmenting the local model update with global generator (see Fig. 1 (a)), and guiding the local generator update with trained local model (see Fig. 1 (b)).

**Augmenting the local model update with global generator.** In the classical local model update, client $i$ leverages the following classification loss to optimize the local model $\boldsymbol{\theta}^i = [\boldsymbol{\theta}_F^i, \boldsymbol{\theta}_D^i]$:

$$\mathcal{L}_{ce,i} = CE(\rho(D_i(F_i(\boldsymbol{x}))), y), \tag{1}$$

where $\rho$ is the softmax function and $CE$ is the cross-entropy function. However, $\mathcal{L}_{ce,i}$ has no access to global knowledge in our work, which is embedded in the global generator. To transfer the knowledge of the global generator to the local model efficiently, we construct the following two losses based on KD:

$$\overrightarrow{\mathcal{L}}_{mse,i} = \|F_i(\boldsymbol{x}) - G(\hat{\boldsymbol{z}}, \boldsymbol{y})\|^2, \overrightarrow{\mathcal{L}}_{kl,i} = KL(\rho(D_i(F_i(\boldsymbol{x})))\|\rho(D_i(G(\hat{\boldsymbol{z}}, \boldsymbol{y})))), \tag{2}$$

where $\hat{\boldsymbol{z}}$ is sampled from $\mathcal{N}(\boldsymbol{0}, \boldsymbol{I})$. $\|\cdot\|^2$ and $KL$ are $L_2$-norm function and Kullback-Leibler function, respectively. Specifically, $\overrightarrow{\mathcal{L}}_{mse,i}$ utilizes $L_2$-norm function to enforce the output of feature extractor $F_i(\boldsymbol{x})$ to approximate that of global generator $G(\hat{\boldsymbol{z}}, \boldsymbol{y})$. After that, client $i$ feeds both $F_i(\boldsymbol{x})$ and $G(\hat{\boldsymbol{z}}, \boldsymbol{y})$ into the classifier to get $D_i(F_i(\boldsymbol{x}))$ and $D_i(G(\hat{\boldsymbol{z}}, \boldsymbol{y}))$. Further, $\overrightarrow{\mathcal{L}}_{kl,i}$ harnesses Kullback-Leibler function to make $D_i(F_i(\boldsymbol{x}))$ close to $D_i(G(\hat{\boldsymbol{z}}, \boldsymbol{y}))$.

To further augment the local model update, client $i$ resamples a batch of noisy data to feed the generator and classifier sequentially, and minimizes the following classification loss:

$$\overrightarrow{\mathcal{L}}_{ce,i} = CE(\rho(D_i(G(\hat{\boldsymbol{z}}, \hat{\boldsymbol{y}}))), \hat{y}), \tag{3}$$

where $\hat{y} \sim p(y) \propto \sum_{i \in [N]} n_i^y$, $n_i^y$ denotes the number of samples w.r.t class $y$ on the $i$-th client.

Combining $\mathcal{L}_{ce,i}$, $\overrightarrow{\mathcal{L}}_{ce,i}$, $\overrightarrow{\mathcal{L}}_{mse,i}$ and $\overrightarrow{\mathcal{L}}_{kl,i}$, the overall objective of the local model update can be formalized as follows:

$$\min_{\boldsymbol{\theta}_F^i, \boldsymbol{\theta}_D^i} \mathbb{E}_{\substack{\hat{\boldsymbol{z}}, \hat{y} \sim \mathcal{N}(\boldsymbol{0}, \boldsymbol{I}), p(y) \\ \boldsymbol{x}, y \sim \boldsymbol{X}_i, \boldsymbol{Y}_i}} [\mathcal{L}_{ce,i} + \lambda_1 \overrightarrow{\mathcal{L}}_{ce,i} + \lambda_2 \overrightarrow{\mathcal{L}}_{mse,i} + \lambda_3 \overrightarrow{\mathcal{L}}_{kl,i}], \tag{4}$$

where $\lambda_1$, $\lambda_2$ and $\lambda_3$ are tunable hyperparameters for balancing different loss items.

**Guiding the local generator update with trained local model.** After the local model update, we maintain a local generator in client $i$ to extract the knowledge of the trained local model without accessing its private data. Note that the global generator does not replace the local generator in our work to learn the knowledge of the trained local model.

Similar to the manner of augmenting local model update, we utilize KD to construct losses $\overleftarrow{\mathcal{L}}_{kl,i}$ and $\overleftarrow{\mathcal{L}}_{mse,i}$ to transfer the knowledge of the local model to the local generator. $\overleftarrow{\mathcal{L}}_{kl,i}$ and $\overleftarrow{\mathcal{L}}_{mse,i}$ take the

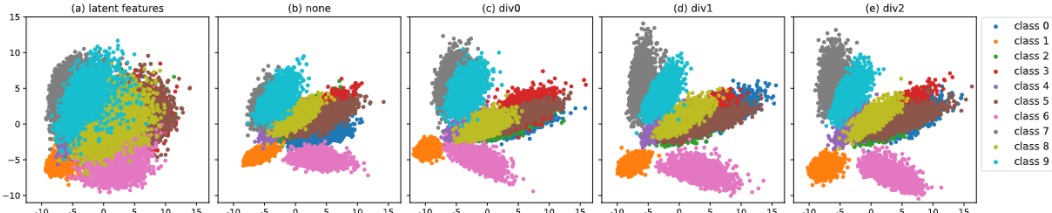

Figure 2: Visualization for output of the generator: The toy example first trains a LeNet (LeCun et al., 1998) as teacher model (T) using the training set of MNIST (LeCun et al., 1998). Then the test set of MNIST is fed to T to get the latent features. And the dimensions of the latent features are reduced by principal component analysis (PCA) (Halko et al., 2011). (a) shows the latent features distribution of T after PCA dimension reduction. Next, we let T guide the training of the generator according to Eq. (9). Similarly, we utilize PCA to perform dimension reduction for the output of the generator. (b) visualizes the output distribution of the generator without diversity constraint. (c), (d) and (e) visualize the output distribution of the generator with $\mathcal{L}_{div}^0$, $\mathcal{L}_{div}^1$ and $\mathcal{L}_{div}^2$, respectively.

forms:

$$\overleftarrow{\mathcal{L}}_{mse,i} = \|G_i(\hat{\boldsymbol{z}}, \boldsymbol{y}) - F_i(\boldsymbol{x})\|^2, \overleftarrow{\mathcal{L}}_{kl,i} = KL(\rho(D_i(G_i(\hat{\boldsymbol{z}}, \boldsymbol{y}))) \| \rho(D_i(F_i(\boldsymbol{x})))). \tag{5}$$

To ensure the fidelity of the output of the local generator $G_i$, $G_i$ is expected to fit the input space of the local classifier for better knowledge extraction from the local model. Therefore, client $i$ takes the following classification loss $\overleftarrow{\mathcal{L}}_{ce,i}$ to enforce $G_i$ to yield higher prediction on class $y$:

$$\overleftarrow{\mathcal{L}}_{ce,i} = CE(\rho(D_i(G_i(\hat{\boldsymbol{z}}, \boldsymbol{y}))), y). \tag{6}$$

However, if we only optimize $\overleftarrow{\mathcal{L}}_{kl,i}$, $\overleftarrow{\mathcal{L}}_{mse,i}$ and $\overleftarrow{\mathcal{L}}_{ce,i}$ for $G_i$, it is likely to generate similar outputs for each class with little diversity, which can cause the model collapse of the local generator. To tackle this limitation, the constraint $\mathcal{L}_{div}^0$ has been added to enhance the output diversity of the generator as follows (Yoo et al., 2019; Zhu et al., 2021; Zhang et al., 2022c):

$$\mathcal{L}_{div}^0 = e^{\frac{1}{B^2} \sum\limits_{j,k \in [B]} \left( -\|\hat{\boldsymbol{f}}_j - \hat{\boldsymbol{f}}_k\|_2 * \|\hat{\boldsymbol{z}}_j - \hat{\boldsymbol{z}}_k\|_2 \right)}, \tag{7}$$

where $B$ denotes the batch size and $\hat{\boldsymbol{f}}_{j/k} = G_i(\hat{\boldsymbol{z}}_{j/k}, \boldsymbol{y}_{j/k})$. This constraint treats the noise pair distance $\|\hat{\boldsymbol{z}}_j - \hat{\boldsymbol{z}}_k\|_2$ as a weight, which is then multiplied by the corresponding output pair distance $\|\hat{\boldsymbol{f}}_j - \hat{\boldsymbol{f}}_k\|_2$ in each batch $B$, thus imposing more weights on the output pairs whose corresponding noise pairs are more distant. It can be found that this weighting scheme of $\mathcal{L}_{div}^0$ is label-agnostic. In other words, the weight differences between intra- and inter-class output pairs are not considered in $\mathcal{L}_{div}^0$, which may lead to inter-class output pairs being close but intra-class output pairs being distant, thus adversely affecting the performance of $G_i$. To rectify this issue, we propose two novel diversity constraints, $\mathcal{L}_{div}^1$ and $\mathcal{L}_{div}^2$. In terms of $\mathcal{L}_{div}^1$, we simply replace $\|\hat{\boldsymbol{z}}_j - \hat{\boldsymbol{z}}_k\|_2$ of Eq. (7) with $\|\hat{\boldsymbol{z}}_j^y - \hat{\boldsymbol{z}}_k^y\|_2$, where $\hat{\boldsymbol{z}}_j^y = [\hat{\boldsymbol{z}}_j; \boldsymbol{y}_j]$. Further, we formulate $\mathcal{L}_{div}^2$ in the following form:

$$\mathcal{L}_{div}^2 = e^{\frac{1}{B^2} \sum\limits_{j,k \in [B]} \left( -\|\hat{\boldsymbol{f}}_j - \hat{\boldsymbol{f}}_k\|_2 * \|\hat{\boldsymbol{z}}_j - \hat{\boldsymbol{z}}_k\|_2 * e^{\|\hat{\boldsymbol{y}}_j - \hat{\boldsymbol{y}}_k\|_1} \right)}. \tag{8}$$

Compared to $\mathcal{L}_{div}^0$, $\mathcal{L}_{div}^1$ and $\mathcal{L}_{div}^2$ further differentiate the weights of the generator's intra- and inter-class output pair distances, with more weights applied to the inter-class output pair distances. For brevity, we uniformly denote $\mathcal{L}_{div}^0$, $\mathcal{L}_{div}^1$ and $\mathcal{L}_{div}^2$ as $\mathcal{L}_{div}$ unless otherwise noted. In Fig. 2, we provide a toy example showing the output distribution of the generator without diversity constraints as well as with different diversity constraints.

We combine $\overleftarrow{\mathcal{L}}_{kl,i}$, $\overleftarrow{\mathcal{L}}_{mse,i}$, $\overleftarrow{\mathcal{L}}_{ce,i}$ and $\mathcal{L}_{div}$ to yield the overall objective of the local generator update for client $i$ is shown below:

$$\min_{\boldsymbol{w}_i} \mathbb{E}_{\substack{\hat{\boldsymbol{z}} \sim \mathcal{N}(\boldsymbol{0}, \boldsymbol{I}) \\ \boldsymbol{x}, y \sim \boldsymbol{X}_i, \boldsymbol{Y}_i}} [\overleftarrow{\mathcal{L}}_{kl,i} + \lambda_4 \overleftarrow{\mathcal{L}}_{mse,i} + \lambda_5 \overleftarrow{\mathcal{L}}_{ce,i} + \lambda_6 \mathcal{L}_{div,i}], \tag{9}$$

where $\lambda_4$, $\lambda_5$ and $\lambda_6$ are non-negative hyperparameters. $\mathcal{L}_{div,i}$ denotes the diversity constraint of client $i$.

## 2.2 Server-side Crossed Distillation Aggregation

After gathering local generators and classifiers uploaded by clients, the server aggregates them as a preliminary global generator and classifier via weighted averaging. However, straightforward average aggregation may counteract the local knowledge from clients. To alleviate this issue, we train the preliminary global generator and classifier via crossed data-free KD to distill as much knowledge as possible from the local generators and classifiers. Fig. 1 (c) shows the distillation schema on the server, where the overall distillation objective consists of three parts: $\mathcal{L}_{kl,1}$, $\mathcal{L}_{kl,2}$ and $\mathcal{L}_{kl,3}$.

Specifically, the server first samples $(\hat{z}, \hat{y})$, and feeds it to the local generators $\{G_i\}_{i \in [N]}$ and the global generator $G$, where $\hat{z} \sim \mathcal{N}(\mathbf{0}, \mathbf{I})$ and $\hat{y} \sim p(y) \propto \sum_{i \in [N]} n_i^y$. Their outputs are then fed into the corresponding classifiers to compute the loss $\mathcal{L}_{kl,1}$:

$$\mathcal{L}_{kl,1} = \sum_{i \in [N]} \tau_{i,\hat{y}} KL(\rho_g \| \rho_i), \tag{10}$$

where $\rho_g = \rho(D(G(\hat{z}, \hat{y})))$, $\rho_i = \rho(D_i(G_i(\hat{z}, \hat{y})))$, and $\tau_{i,\hat{y}} = n_i^{\hat{y}} / \sum_{j \in [N]} n_j^{\hat{y}}$. $\mathcal{L}_{kl,1}$ ensures that the $\rho_g$ from the global classifier approximates $\{\rho_i\}_{i \in [N]}$ from the local classifiers. However, simply distilling knowledge by minimizing $\mathcal{L}_{kl,1}$ could be insufficient, since $D_i$ fits $G_i(\hat{z}, \hat{y})$ (via optimizing Eq. (9)) but may not fit $G(\hat{z}, \hat{y})$ and $G_i(\hat{z}, \hat{y})$ fits $D_i$ but may not fit $D$, such that only partial knowledge from clients can be extracted. Therefore, to address these limitations, we introduce a crossover strategy and formulate two losses $\mathcal{L}_{kl,2}$ and $\mathcal{L}_{kl,3}$ as:

$$\mathcal{L}_{kl,2} = \sum_{i \in [N]} \tau_{i,\hat{y}} KL(\rho_{ig} \| \rho_i), \mathcal{L}_{kl,3} = \sum_{i \in [N]} \tau_{i,\hat{y}} KL(\rho_{gi} \| \rho_i), \tag{11}$$

where $\rho_{ig} = \rho(D(G_i(\hat{z}, \hat{y})))$ and $\rho_{gi} = \rho(D_i(G(\hat{z}, \hat{y})))$.

Further, $\mathcal{L}_{kl,1}$, $\mathcal{L}_{kl,2}$ and $\mathcal{L}_{kl,3}$ form the following overall distillation objective on the server side:

$$\min_{\boldsymbol{w}, \boldsymbol{\theta}_D} \mathbb{E}_{\hat{z}, \hat{y} \sim \mathcal{N}(\mathbf{0}, \mathbf{I}), p(y)} [\mathcal{L}_{kl,1} + \mathcal{L}_{kl,2} + \mathcal{L}_{kl,3}]. \tag{12}$$

## 2.3 Discussion

**Privacy.** FedMD-CG trains a local generator on each client for replacing the local feature extractor (LFE) by simulating the output vector space of LFE, i.e., the latent feature space, rather than the distribution space of private data. In other words, the local generator captures only high-level feature patterns of the local model, which are incomprehensible to human beings. Also, FedMD-CG requires each client to share its classifier. In our work, the classifier is in the top layers (i.e., fully connected layers) with a high degree of abstraction. As verified by (Yosinski et al., 2014), the lower layer features are more general and higher layer features have larger specificity. This suggests that different inputs to the model can result in the same top-layer activations, making it difficult to reconstruct the original data with the classifier (Wang, 2021). Therefore, FedMD-CG can reduce the risk of privacy leakage, and has the same level of privacy protection as FedCG.

**Consistency and Computing cost.** FedMD-CG performs client-side knowledge transfer at the latent feature level and the logit level, thus extracting knowledge embedded in the global generator and local model more directly and efficiently than FedCG. Meanwhile, our method guarantees the consistency of the local generator and classifier trained by each client, which may not be satisfied in FedCG. To put it differently, FedMD-CG requires the local generator to generate pseudo-features that the local classifier can significantly distinguish in order to make the generator output more fidelity. According to our experiments, the consistency of FedMD-CG ensures high-quality aggregation on the server side and robustness to data heterogeneity. In addition, FedMD-CG does not employ an additional discriminator to adversarially train the local generator under the cGAN framework independently of the local classifier like FedCG, which reduces the client's computing cost.

Table 1: Test performance (%) comparison between FedMD-CG and baselines over different datasets. Note that $L.acc$ and $G.acc$ denote *local test accuracy* and *global test accuracy*, respectively.

| | EMNIST | | | | FMNIST | | | | CIFAR-10 | | | |
|---|---|---|---|---|---|---|---|---|---|---|---|---|
| | $\omega = 1.0$ | | $\omega = 0.1$ | | $\omega = 1.0$ | | $\omega = 0.1$ | | $\omega = 10.0$ | | $\omega = 1.0$ | |
| Alg.s | $L.acc$ | $G.acc$ | $L.acc$ | $G.acc$ | $L.acc$ | $G.acc$ | $L.acc$ | $G.acc$ | $L.acc$ | $G.acc$ | $L.acc$ | $G.acc$ |
| FedAvg | **96.29±0.06** | 96.83±0.07 | **85.65±2.33** | **95.06±0.46** | **80.99±0.81** | **84.77±0.30** | 59.29±3.19 | **78.91±2.12** | 51.68±0.53 | 54.18±0.52 | 42.84±2.03 | **52.39±0.58** |
| LT | 90.48±1.46 | 95.17±0.70 | 40.77±2.34 | 62.08±5.11 | 74.19±2.92 | 80.32±1.02 | 37.71±2.99 | 56.34±10.42 | 49.83±0.88 | 48.99±1.35 | 40.92±2.25 | 37.92±2.24 |
| FedPer | 91.96±1.20 | 96.45±0.10 | 41.08±2.40 | 76.90±2.34 | 75.83±2.42 | 82.94±1.11 | 37.39±3.17 | 61.88±9.80 | 50.72±0.63 | 53.87±0.58 | 41.80±2.15 | 49.83±1.66 |
| LG-FedAvg | 94.01±0.53 | 96.22±0.19 | 46.29±3.39 | 86.48±1.75 | 77.03±1.94 | 82.55±0.46 | 38.89±3.18 | 66.35±6.65 | 50.11±0.80 | 51.80±0.67 | 41.49±2.56 | 44.59±1.88 |
| FedGen | 95.62±0.38 | 97.64±0.17 | 51.29±4.01 | 87.69±2.50 | 77.88±3.10 | 83.81±1.95 | 41.96±3.40 | 68.05±3.96 | 52.94±2.38 | 48.49±3.13 | 38.13±4.89 | 40.85±3.87 |
| FedCG | 96.06±0.33 | **97.70±0.16** | 49.91±3.83 | 87.66±2.08 | 74.92±2.11 | 81.74±0.81 | 34.97±2.55 | 54.61±2.67 | 39.39±5.23 | 37.06±4.35 | 30.44±3.30 | 26.79±2.82 |
| FedMD-CG | 95.45±0.25 | 97.18±0.17 | 54.45±3.56 | 87.87±1.64 | 79.00±1.43 | 84.47±0.38 | 42.55±3.68 | 71.09±1.01 | **54.82±0.79** | **55.18±1.75** | **46.30±2.24** | 47.56±2.21 |

## 3 EXPERIMENTS

### 3.1 IMPLEMENTATION SETTINGS

**Datasets.** We perform our experiments on three public datasets EMNIST (Cohen et al., 2017), Fashion-MNIST (Xiao et al., 2017) (FMNIST in short in this paper), and CIFAR-10 (Krizhevsky et al., 2009). Following existing works (Zhang et al., 2022c; Acar et al., 2021; Zhu et al., 2021), we use Dirichlet process $Dp(\omega)$ to strictly partition the training set of each dataset across clients. Notably, a smaller $\omega$ corresponds to higher data heterogeneity. We set $\omega \in \{0.1, 1.0, 10.0\}$ in our experiments.

**Backbone Architectures and Baselines.** Throughout all our experiments, we deploy LeNet-5 (LeCun et al., 1998) as the backbone network with two convolutional layers (i.e., feature extractor) and three fully connected layers (i.e., classifier). Similarly, we employ three fully connected layers with BatchNorm as the generator for each client and adjust its output dimension to match that of the corresponding feature extractor. We select five FL methods most relevant to our work as baselines for comparison, including FedAvg (McMahan et al., 2017), FedPer (Arivazhagan et al., 2019), LG-FedAvg (Liang et al., 2020), FedGen[1] (Zhu et al., 2021) and FedCG (Wu et al., 2021). Moreover, we consider the baseline that trains a local model for each client, without any sharing. We call it Local Training (LT for short). For fairness, FedGen shares clients' classifiers with the server. In particular, we treat the client's classifier whose output dimension is set to 1 as the discriminator of cGAN in FedCG.

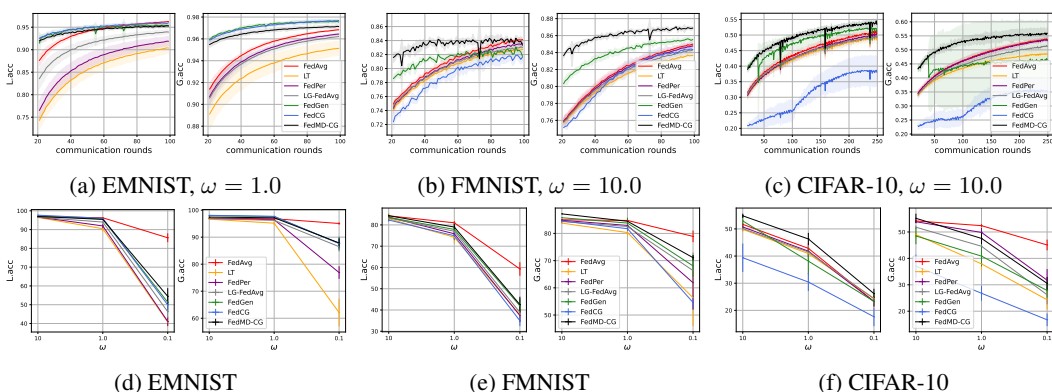

(a) EMNIST, $\omega = 1.0$  (b) FMNIST, $\omega = 10.0$  (c) CIFAR-10, $\omega = 10.0$

(d) EMNIST  (e) FMNIST  (f) CIFAR-10

Figure 3: (a)-(c) are learning curves selected from FedMD-CG as well as baselines over different datasets. (d)-(f) show test performance (%) w.r.t data hetergeneity over each dataset.

**Evaluation Metrics.** We use the test set of each dataset to evaluate the test performance of different FL methods. 1) *Local test accuracy*. We randomly and evenly distribute the test set to each client and harness the test set on each client to verify the performance of local models. 2) *Global test accuracy*. We construct a virtual global model to evaluate the global performance of different FL methods via utilizing the original test set. As with FedAvg, this virtual global model is obtained by uploading all local models to the server for weighted average. 3) *Peak signal-to-noise ratio (PSNR)*. Consistent

---

[1]In this paper, we consider FedGen with partial parameter sharing.

with FedCG (Wu et al., 2021), we also consider the server is malicious, which uses DLG attack (Zhu et al., 2019) to recover the original data from victim clients. We employ PSNR to measure the quality of the recovered images, thus evaluating the privacy-preserving capability of different FL methods. To ensure reliability, we report the average for each experiment over 5 different random seeds. Due to the space limitations, we relegate full experimental settings and results to Appendix E.

## 3.2 RESULTS COMPARISON

**Overview test performance comparison.** As shown in Table 1, FedAvg achieves the best test performance while FedMD-CG achieves the second-best test performance in most cases of EMNIST and FMNIST. FedAvg's test performance benefits from the fact that the server can collect complete local models from clients and then obtain the real global model to ensure remarkable test performance. In most cases, the test performance of LT is worse than that of other methods since no information is shared among clients, inevitably causing over-fitting and poor generalization to new samples. LG-FedAvg consistently out-performs FedPer w.r.t the local test accuracy,

Table 2: Comparison of FedMD-CG and baselines in terms of PSNR (dB) ($\omega = 10.0$). Note that both FedGen and LG-FedAvg upload local classifiers to the server, and their privacy-preserving capabilities are intuitively the same, so we only report the PSNR of LG-FedAvg.

| Alg.s | EMNIST | FMNIST | CIFAR-10 |
|---|---|---|---|
| FedAvg | 24.54±0.15 | 22.63±0.61 | 29.22±1.56 |
| FedPer | 23.55±0.52 | 19.66±1.03 | 14.84±2.61 |
| LG-FedAvg | **6.78±0.09** | **6.33±1.32** | **8.75±1.03** |
| FedCG | 7.05±0.63 | 6.98±1.57 | 9.87±1.83 |
| FedMD-CG | 6.95±0.31 | 7.02±1.22 | 9.69±1.04 |

indicating that personalized classifiers can mitigate the sacrifice of local model performance when the feature extractor has several convolutional layers. Meanwhile, FedMD-CG achieves the optimal local test accuracy on CIFAR-10. We conjecture that the simple model average aggregation in FedAvg may counteract the personalized knowledge from clients, thus adversely affecting local models' performance in difficult classification tasks. Further, Fig. 3 (b)-(e) demonstrate that there is an over-whelming advantage of FedMD-CG over baselines in terms of learning efficiency during the early stages of training. Particularly, the local learning efficiency of FedMD-CG consistently outperforms that of baselines on FMNIST with $\omega = 10.0$ and CIFAR-10. Fig. 3 (f)-(h) reveal the impact of data heterogeneity on test performance for the methods. It can be observed that the test performance of all methods deteriorates as $\omega$ decreases. In most cases, FedMD-CG dominates the baselines that share only part of the model in terms of the local test accuracy. Also, FedMD-CG uniformly surpasses baselines w.r.t the local test accuracy over varying $\omega$ on CIFAR-10. This indicates that our method is robust to data heterogeneity.

**Privacy comparison.** Here, we compare the privacy-preserving ability of FedMD-CG with other baselines under DLG attack. It is worth noting that PSNR measures the similarity be-tween the original image and the restored im-age. A larger PSNR value indicates a higher similarity between the images. As observed in Table 2, while FedAvg achieves excellent test performance (see Table 1), it scores the high-est PSNR value across all datasets, which se-riously threatens clients' private information. Also, Fig. 4 illustrates that the DLG attack is able to reconstruct the image very close to the original image in FedAvg. According to Table 1 and Fig. 4, it is noticed that the strategy in Fed-Per to share clients' feature extractors should be prohibited, as it neither enables competitive test performance nor protects clients' privacy. On

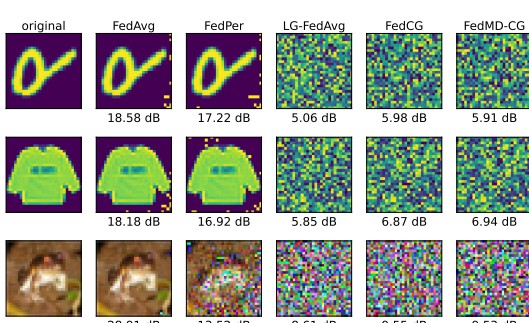

Figure 4: Image reconstruction with DLG attack in FedMD-CG and baselines. From the first to the last row, the images are selected from EMNIST, FMNIST and CIFAR-10 respectively. PSNR (dB) is reported under each recovered image.

the other hand, LG-FedAvg, FedCG and FedMD-CG can effectively prevent the privacy leakage of clients due to the low PSNR values. Despite the small performance gap between FedMD-CG and LG-FedAvg w.r.t. PSNR, FedMD-CG can significantly outperform LG-FedAvg in terms of

Table 3: Test performance (%) comparison FedMD-CG and FedCG with different server-side aggregation manners over different datasets. Note that *AVE_agg* and *AVE_agg*⋆ denote weighted average aggregation. Specifically, FedMD-CG with *AVE_agg* transfers the knowledge from the global generator to local models at both the latent feature level and the logit level, whereas FedMD-CG with *AVE_agg*⋆ transfers the knowledge from the global generator to local models only at the latent feature level. Also, *KD_agg* and *KDC_agg* denote the server-side aggregation manners from FedCG and FedMD-CG, respectively.

| Alg.s | Agg. | EMNIST, $\omega = 0.1$ | | FMNIST, $\omega = 0.1$ | | CIFAR-10, $\omega = 1.0$ | |
|---|---|---|---|---|---|---|---|
| | | *L.acc* | *G.acc* | *L.acc* | *G.acc* | *L.acc* | *G.acc* |
| FedCG | *AVE_agg*⋆ | 50.55±4.32 | 86.74±1.44 | 39.46±3.40 | 67.41±3.59 | 41.85±2.48 | 44.98±1.79 |
| | *KD_agg* | 49.91±3.83 | 87.66±2.08 | 34.97±2.55 | 54.61±2.67 | 30.44±3.30 | 26.79±2.82 |
| | *KDC_agg* | 39.65±4.67 | 82.32±4.66 | 37.23±2.54 | 62.89±7.01 | 28.87±0.90 | 25.92±1.53 |
| FedMD-CG | *AVE_agg*⋆ | 51.36±3.63 | 86.88±1.53 | 40.55±3.55 | 67.34±5.41 | 43.24±2.32 | 45.10±1.44 |
| | *AVE_agg* | 52.62±3.74 | 86.92±1.35 | 41.44±2.98 | 67.88±6.07 | 45.16±2.35 | 46.72±2.32 |
| | *KD_agg* | 53.14±4.73 | 83.86±2.10 | 41.79±3.54 | 64.68±4.31 | 45.12±2.30 | 46.98±2.60 |
| | *KDC_agg* | **54.45±3.56** | **87.87±1.64** | **42.55±3.68** | **71.09±1.01** | **46.30±2.24** | **47.56±2.21** |

test performance (see Table 1). This indicates that approximating the local feature extractor with a generator not only has little privacy leakage risk but also improves performance.

**Comparison between FedCG and FedMD-CG.** From Table 1 and Fig. 3, the test performance of FedCG is worse than that of FedMD-CG in most cases, even worse than other baselines on CIFAR-10 and FMNIST ($\omega = 0.1$). We speculate that this attributes to the way FedCG transfers knowledge from the global generator to the local model and the inconsistency between the local generator and classifier in each client. We next perform extensive experiments to verify our statement, as shown in Table 3 and Fig. 5. From Table 3, FedMD-CG with *AVE_agg*⋆ consistently surpasses FedCG with *AVE_agg*⋆ in terms of the local test accuracy. The main reason is that FedMD-CG enables the local generators

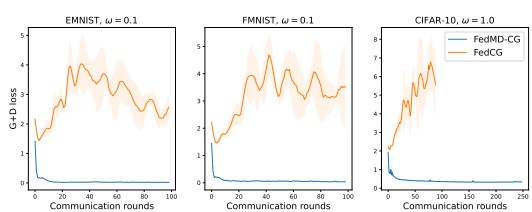

Figure 5: The consistency comparison between local generators and classifiers for FedCG and FedMD-CG w.r.t. *AVE_agg*⋆. G+D loss denotes the classification loss of the local classifier on the output of the local generator.

to extract the knowledge of the local models more effectively, which results in higher-quality trained local generators. Also, the test performance of FedMD-CG with *AVE_agg* uniformly leads that of FedMD-CG with *AVE_agg*⋆, suggesting the insufficiency of knowledge transfer from the global generator to local models only at the latent feature level. In addition, we compare the efficacy of different server-side aggregation manners. Concretely, the test performance of FedMD-CG with *KDC_agg* consistently outperforms FedMD-CG with *KD_agg*, indicating that *KDC_agg* is more effective in transferring knowledge from local generators and classifiers to the global generator and classifier. However, *KD_agg* and *KDC_agg* significantly deteriorate the test performance of FedCG. We conjecture that the output of the local generator does not match the local classifier's, that is, the local classifier cannot effectively distinguish the output of the local generator, resulting in the degraded performance of FedCG. As shown in Fig. 5, G+D loss of FedCG is consistently larger than that of FedMD-CG and does not converge. In general, the inconsistency of the local generator and classifier in each client may impede the server-side knowledge distillation aggregation training, resulting in poor performance of FedCG.

## 3.3 ABLATION STUDY

**Necessity of losses in client-side for FedMD-CG.** We look into the test performance of FedMD-CG on CIFAR-10 with $\omega = 10.0$ after discarding some losses in Eqs. (4) and (9), respectively, as shown in Table 4. We can see that removing any loss leads to worse performance, i.e., lower local test accuracy and global test accuracy. Also, their joint absence can cause further degradation of test performance. A trend in losses is observed that the absence of a single loss leads to a drop in test performance, while the removal of multiple losses enlarges the drop. In addition, it should be noted that dropping multiple losses in the local generator update leads to more severe test performance degradation compared to the local model update. This shows that well-trained local generators can

Table 4: Impact of each loss for client-side training over CIFAR-10 with $\omega = 10.0$. Note that L.M.U and L.G.U denote the local model update and the local generator update, respectively. Also, we omit the subscript $i$ of each loss for client $i$.

| FedMD-CG (baseline) | | | | | |
|---|---|---|---|---|---|
| L.acc | | | G.acc | | |
| 54.82±0.79 | | | 55.18±1.75 | | |
| L.M.U | L.acc | G.acc | L.G.U | L.acc | G.acc |
| $-\vec{\mathcal{L}}_{ce}$ | 51.53±1.04 | 52.52±1.37 | $-\overleftarrow{\mathcal{L}}_{mse}$ | 52.73±1.15 | 53.19±1.72 |
| $-\vec{\mathcal{L}}_{mse}$ | 53.07±0.97 | 53.73±1.99 | $-\overleftarrow{\mathcal{L}}_{ce}$ | 53.89±0.89 | 52.88±2.09 |
| $-\vec{\mathcal{L}}_{kl}$ | 53.46±0.94 | 53.34±1.74 | $-\mathcal{L}_{div}$ | 52.66±0.77 | 53.11±1.72 |
| $-\vec{\mathcal{L}}_{ce}, -\vec{\mathcal{L}}_{mse}$ | 51.02±0.52 | 52.96±1.01 | $-\overleftarrow{\mathcal{L}}_{mse}, -\overleftarrow{\mathcal{L}}_{ce}$ | 46.94±1.33 | 49.37±1.53 |
| $-\vec{\mathcal{L}}_{ce}, -\vec{\mathcal{L}}_{kl}$ | 51.55±0.60 | 52.81±1.22 | $-\overleftarrow{\mathcal{L}}_{mse}, -\mathcal{L}_{div}$ | 47.80±0.30 | 50.15±1.24 |
| $-\vec{\mathcal{L}}_{mse}, -\vec{\mathcal{L}}_{kl}$ | 52.64±0.40 | 53.46±1.25 | $-\overleftarrow{\mathcal{L}}_{ce}, -\mathcal{L}_{div}$ | 48.02±0.31 | 50.41±1.61 |
| $-\vec{\mathcal{L}}_{ce}, -\vec{\mathcal{L}}_{mse}, -\vec{\mathcal{L}}_{kl}$ | 50.27±0.41 | 49.55±1.28 | $-\overleftarrow{\mathcal{L}}_{mse}, -\overleftarrow{\mathcal{L}}_{ce}, -\mathcal{L}_{div}$ | 44.33±1.38 | 47.66±1.35 |

effectively boost the performance of our method, while under-trained local generators hinder the training of models.

**Impacts of diversity constraints.** We also explore the effect of different diversity constraints on FedMD-CG. Note that we omit the subscript $i$ of diversity loss for client $i$. From Table 5, FedMD-CG with $\mathcal{L}_{div}^1$ and $\mathcal{L}_{div}^2$ beats FedMD-CG with $\mathcal{L}_{div}^0$ w.r.t. the test performance in most case. Also, $\mathcal{L}_{div}^1$ and $\mathcal{L}_{div}^2$ uniformly trump $\mathcal{L}_{div}^0$ in terms of the local test accuracy. Consequently, an empirical finding can be derived that imposing more weight on the inter-class output pair distance of the local generator boosts the local models' performance.

Table 5: Test performance (%) comparison among different diversity constraints.

| Div. con. | EMNIST, $\omega = 0.1$ | | FMNIST, $\omega = 1.0$ | | CIFAR-10, $\omega = 10.0$ | |
|---|---|---|---|---|---|---|
| | L.acc | G.acc | L.acc | G.acc | L.acc | G.acc |
| $\mathcal{L}_{div}^0$ | 53.09±4.27 | **88.85±1.31** | 78.58±1.58 | 84.69±0.46 | 54.24±0.72 | 54.78±1.88 |
| $\mathcal{L}_{div}^1$ | 53.65±4.12 | 88.51±0.80 | **79.03±1.52** | **84.73±0.49** | 54.81±0.71 | 54.90±1.73 |
| $\mathcal{L}_{div}^2$ | **54.45±3.56** | 87.87±1.64 | 79.00±1.43 | 84.47±0.38 | **54.82±0.79** | **55.18±1.75** |

**Robustness of FedMD-CG against hyperparameters.** We investigate the test performance of FedMD-CG with varying hyperparameters over FMNIST. We set $\omega = 1.0$ and select $\lambda_1$, $\lambda_2$, $\lambda_3$, $\lambda_4$, $\lambda_5$ and $\lambda_6$ from $[0.25, 0.5, 0.75, 1.0, 1.25, 1.5]$. Fig. 6 shows the test performance using the box plot, where FedMD-CG exemplifies similar test performance for non-zero selection of hyperparameters. Notably, for a single loss, the effect of non-zero varying hyperparameters on the local test accuracy of FedMD-CG is slight. This indicates that FedMD-CG is insensitive to the choice of non-zero hyperparameters over a large range for a single loss.

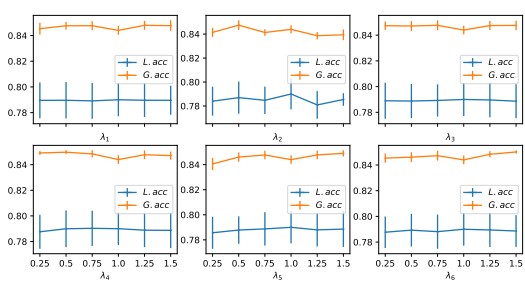

Figure 6: Test performance of FedMD-CG using varying hyperparameters on FMNIST with $\omega = 1.0$.

# 4 CONCLUSIONS

In this paper, we propose a novel FL method FedMD-CG, which achieves high competitive performance and high-level privacy preservation. Specifically, FedMD-CG decomposes each client's local model into a feature extractor and a classifier, and utilizes a conditional generator instead of the feature extractor to perform server-side model aggregation. Meanwhile, our method taps KD to train local models and generators at the latent feature level and the logit level, thereby ensuring the consistency of local generators and classifiers. Also, we construct additional classification losses and craft new diversity losses to enhance client-side training. On the server side, FedMD-CG aggregates trained local generators and classifiers in a crossed data-free KD manner. Finally, we conduct extensive experiments to verify the superiority of FedMD-CG. Due to space constraints, we discuss in detail the **limitations** and **broader impacts** of our work in Appendixes F and G, respectively.

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
