# OpenReview forum: "Model-Decoupling-Based Federated Learning with Consistency via Knowledge Distillation Using Conditional Generator"
_ICLR.cc/2024/Conference — ICLR 2024 Conference Withdrawn Submission_

### Official Review · Reviewer_eFsa · 2023-10-29

**Soundness:** 2 fair
**Presentation:** 2 fair
**Contribution:** 2 fair
**Rating:** 3
**Confidence:** 4

**Summary:**

The paper proposes a novel FL algorithm, FedMD-CG, based on knowledge-distillation, which enabling clients to train a local generator for extracting the local knowledge and the server to aggregate the local knowledge from clients accordingly. Since FedMD-CG does not communicate the local feature extractors to the server, it provides high-level of protection for clients under gradient inversion attack. Extensive experimental results validate the effect of  FedMD-CG in accuracy and privacy.

**Strengths:**

1. The paper attempts to explore an interesting area: data-free knowledge distillation in Federated Learning.
2. The proposed method, FedMD-CG, improves both test accuracy as well as privacy-protection in Federated Learning.

**Weaknesses:**

1. For the classification task (which the paper is mainly focusing on), the proposed method requires clients not only to train a model for classification (feature extractor and classifier) as well as a local generator. For the server, it needs to train a global generator. This introduces much more computation overhead. On-edge learning where all clients have limited resources, the propose method might not work. For data-free KD-based FL, there is a prior work that does not need to train a generative model. I think it's better to compare with it [1].

2. The local objective and server's objective functions are so complicated. There are many terms using MSE, KLD and Cross-Entropy Loss. As a result, there are too many hyper-parameters for the regularization terms. However, there is no theoretical analysis on the convergence in the paper. The method is empirical but difficult to reproduce because so many hyper-parameters.

3. The experiments ran on simple datasets, EMNIST, FMNIST,CIFAR-10. The proposed method need to be validated in more larger dataset such as CIFAR100/ImageNet. However, more complicated dataset might need more larger model as generators, which increases the computation overhead.

4. The small problem of the paper: many notations are similar, such as $\mathcal{L}_{mse}$ with $\leftarrow$ and $\rightarrow$, really confusing. And the table 4 in ablation study, those different $\mathcal{L}$ are not straightforward.

Reference:
[1] Chen H, Vikalo H. The Best of Both Worlds: Accurate Global and Personalized Models through Federated Learning with Data-Free Hyper-Knowledge Distillation[J]. arXiv preprint arXiv:2301.08968, 2023.

**Questions:**

1.  what's the insight for the privacy protection? There is no theoretical analysis on the privacy or detailed discussion on why privacy is preserved with the proposed method.

2. how many clients do you run the experiments? If the number of clients is large, is the proposed method still working?

3. The ablation study is running on $\omega = 10$, which can be seen as IID. How about the results in non-IID setting, such as $\omega = 0.1$?

4. Do you have a analysis of comparison overhead? what's the complextiy of your method comparing to FedAvg?

---

> ### Author Response · Authors · 2023-11-14
>
> We highly appreciate Reviewer eFsa for providing thoughtful feedback on our work. We respond to specific comments below.
>
> Q1: For the classification task (which the paper is mainly focusing on), the proposed method requires clients not only to train a model for classification (feature extractor and classifier) as well as a local generator. For the server, it needs to train a global generator. This introduces much more computation overhead. On-edge learning where all clients have limited resources, the propose method might not work. For data-free KD-based FL, there is a prior work that does not need to train a generative model. I think it's better to compare with it [1].
>
> R1: We agree with the reviewer that the proposed method may not work in edge learning where all clients have limited resources.
> As discussed in Appendix F Limitations of our submitted paper, it is extremely challenging to try to develop a universal FL method that can address all problems.
> In our work, we focus on model decoupling (MD)-based FL [2, 3, 4, 5], which has been widely acknowledged to mitigate the risk of privacy leakage of standard FL in the community.  In particular,  our main contribution is to achieve better generalization performance given the same level of privacy protection as FedCG. Additionally, our method significantly outperforms the MD-based baselines in most cases (see Table 1).
>
> In addition, we thank the reviewers for mentioning existing work [1]. We have already studied [1] during our preparation of the manuscript. This work focuses on each client extracts and sends to the server the means of local data representations and the corresponding soft predictions -- information that it refers to as “hyper-knowledge”.
> The server aggregates this information and broadcasts it to the clients in support of local training.
> Also, it promotes privacy with the help of the differential privacy mechanism.
> This can be clearly seen to be orthogonal to our work.
> Therefore, to ensure fair comparisons, we neglect the comparison with the method proposed by [1].
> Besides, we'll add the mentioned paper as one of our references.
>
>
> Q2: The local objective and server's objective functions are so complicated. There are many terms using MSE, KLD and Cross-Entropy Loss. As a result, there are too many hyper-parameters for the regularization terms. However, there is no theoretical analysis on the convergence in the paper. The method is empirical but difficult to reproduce because so many hyper-parameters.
>
> R2: Our approach closely follows the FedCG framework [2]. We propose several incremental innovations to improve upon the FedCG framework.
> Although the proposed changes appear to be complicated, they are simply based on knowledge distillation concepts and most of them are intuitively reasonable.
>
> In addition, we argue that the theoretical guarantee (including privacy protection and convergence) for our method is crucial.
> However, it’s worth noting that even in well-known MD-based federated learning efforts [2, 3, 4, 5], as well as federated learning methods with the help of the generator [2, 3, 6, 7], comprehensive theoretical analysis concerning the privacy guarantees and convergency is often absent. This is also discussed in Appendix 7 Limitations of our submitted paper.
> Given the lack of suitable theoretical frameworks, we concentrated on robust empirical validation, showcasing our method (FedMD-CG).
> Our results, we believe, robustly demonstrate our method’s utility. We intend to delve deeper into theoretical aspects in future work.
>
> Moreover, the empirical results in our paper can all be reproduced by the code we provide.
>
>
> Q3: The experiments ran on simple datasets, EMNIST, FMNIST,CIFAR-10. The proposed method need to be validated in more larger dataset such as CIFAR100/ImageNet. However, more complicated dataset might need more larger model as generators, which increases the computation overhead.
>
> R3: We agree with this point about its application to intricate data and model scenarios. However, all our experiments are conducted on one NVIDIA Tesla V100 GPU with 32Gb of memory, thus running FedMD-CG and baselines on the intricate data and model would incur extreme time costs. Also, we follow existing work FedCG [2] and FedGen [3] to setup backbone architectures (i.e., LeNet-5) in our work. On this account, we concentrated on robust empirical validation, showcasing our method (FedMD-CG).
>
> Q4: The small problem of the paper: many notations are similar, such as $\mathcal{L}_{mse}$ with $\leftarrow$ and $\rightarrow$, really confusing. And the table 4 in ablation study, those different $\mathcal{L}$ are not straightforward.
>
> R4: We will take the reviewers' comments into account and will update them in the revised paper.
>
> Q5: what's the insight for the privacy protection? There is no theoretical analysis on the privacy or detailed discussion on why privacy is preserved with the proposed method.
>
> R5: Please see R2 for details.

---

> ### Author Response · Authors · 2023-11-14
>
> Q6: how many clients do you run the experiments? If the number of clients is large, is the proposed method still working?
>
> R6: Due to the space limitation of the main paper, we detail experimental settings in Appendix E Full Experiments of our submitted paper. Moreover, with sufficient computational resources, our approach still works with a large number of clients.
>
> Q7: The ablation study is running on $\omega = 10$, which can be seen as IID. How about the results in non-IID setting, such as $\omega = 0.1$?
>
> R7: In our work, we have performed comprehensive ablation experiments at EMNIST, FMNIST and CIFAR-10.
> In the ablation study, we focus on the effect of different parts of FedMD-CG on its performance.
> For EMNIST, we set $\omega=0.1$; for FMNIST, we set $\omega=1.0$; for CIFAR-10, we set $\omega=10$.
> Due to the space limitation of the main paper, we only report results about CIFAR-10. See Appendix E.2 of our submitted paper for more results.
>
> Q8: Do you have a analysis of comparison overhead? what's the complextiy of your method comparing to FedAvg?
>
> R8: We provide a discussion of the review's doubts in the second paragraph about Computational Efficiency, Communication Cost and Utility in Appendix F of the submitted paper. Please see Appendix F.
>
> We are honored to address your uncertainties and concerns. We remain dedicated to addressing concerns you may possess with utmost eagerness.
>
> [2] Wu, Y., Kang, Y., Luo, J., He, Y., & Yang, Q. (2021) Fedcg: Leverage conditional gan for protecting privacy and maintaining competitive performance in federated learning. arXiv preprint arXiv:2111.08211.
>
> [3] Zhu, Z., Hong, J., & Zhou, J. (2021) Data-free knowledge distillation for heterogeneous federated learning. In International Conference on Machine Learning, pp. 12878-12889.
>
> [4] Arivazhagan, M. G., Aggarwal, V., Singh, A. K., & Choudhary, S. (2019) Federated learning with personalization layers. arXiv preprint arXiv:1912.00818.
>
> [5] Liang, P. P., Liu, T., Ziyin, L., Allen, N. B., Auerbach, R. P., Brent, D., Salakhutdinov R. & Morency, L. P. (2020) Think locally, act globally: Federated learning with local and global representations. arXiv preprint arXiv:2001.01523.
>
> [6] Lin Zhang, Li Shen, Liang Ding, Dacheng Tao, and Ling-Yu Duan. Fine-tuning global model via data-free knowledge distillation for non-iid federated learning. In CVPR, pp. 10174–10183, 2022c.
>
> [7] Jie Zhang, Chen Chen, Bo Li, Lingjuan Lyu, Shuang Wu, Shouhong Ding, Chunhua Shen, and Chao Wu. Dense: Data-free one-shot federated learning. In Advances in Neural Information Processing Systems, 2022a.

---

> > ### Comment · Reviewer_eFsa · 2023-11-16
> > **Further Discussion on Authors' response**
> >
> > Thank you for the response! Some of my questions have been clarified. Nevertheless, I think the work still has several limitations:
> >
> > (1) According to R2, the authors admit this work is incremental following the framework of FedCG. Yet, the performance of FedMD-CG can't even beat FedCG in some settings.
> >
> > (2) The method can be applied in limited range because of tremendous computation overhead, *''requires clients to have more hardware and computational resources to train generators and local models as compared to FedAvg."* as stated in the paper.
> >
> > (3) The experiments are not comprehensive and confusing. The proposed method does not show dominant performance over other baselines. The simulated systems have only 20 clients and can not verify the performance in larger-scale systems.
> >
> > In summary, this work is not suitable to be published in ICLR at this stage. A lot of limitations need to be solved.

---

> ### Author Response · Authors · 2023-11-19
>
> We greatly appreciate the feedback from the reviewers and the valuable insights provided. We respond to specific comments below.
>
> Q9: According to R2, the authors admit this work is incremental following the framework of FedCG. Yet, the performance of FedMD-CG can't even beat FedCG in some settings.
>
> R9: We acknowledge that in a few cases FedMD-CG does not perform as well as FedCG. e.g., in Table 6 in Appendix E.2, FedCG slightly outperforms FedMD-CG when $\omega =10.0$ and $\omega = 1.0$; however, FedMD-CG significantly beats FedCG in most other cases.
> Our work is indeed inspired by FedCG and improved for its shortcomings.
> We describe this in detail in the third paragraph of Itroduction.
> Specifically, first, knowledge transfer modality at the latent feature level may not be sufficient.
> Second, additional discriminators need to be trained to satisfy the adversarial training of cGAN.
> Third, the trained local generator may not match the local classifier, terming their inconsistency.
> Then, experiments were executed specifically to validate it in the Experimental section, see Table 3.
> Therefore, we believe that our effort is meaningful and can advance the field.
>
>
> Q10: The method can be applied in limited range because of tremendous computation overhead, ''requires clients to have more hardware and computational resources to train generators and local models as compared to FedAvg." as stated in the paper.
>
> R10: We agree with this statement.
> However, our main purpose of using more hardware and computational resources to train the generators is to overcome the drawback that FedAvg is highly vulnerable to inference attacks.
> This is fully verified in Table 2 and Figure 3 in the Experimental section, indicating that FedAvg fail to cope with gradient attacks.
> This is also the aim of the baselines FedPer, LG-FedAvg, FedGen and FedCG, which we have explained in detail in R2.
> As such, the goal of our approach is to guarantee privacy while maximising model performance.
>
> Q11: The experiments are not comprehensive and confusing. The proposed method does not show dominant performance over other baselines. The simulated systems have only 20 clients and can not verify the performance in larger-scale systems.
>
> R11: We argue that we have done comprehensive experiments and provided detailed descriptions to avoid confusion.
> We acknowledge that our method slightly underperformances some methods (e.g., FedCG) in terms of accuracy in a few cases, which we believe is reasonable.
> This is because it is unrealistic and challenging to try to develop a FL method that performs optimally in all cases.
> And our method is clearly superior to state-of-the-art MD-based methods (e.g., FedPer, LG-FedAvg, FedGen, and FedCG) in most cases, see Table 1 and Tables 6-8 in the Appendix E.2.
> Moreover, in the experimental setup part, e.g., the number of clients, we follow the existing works [2, 3] to ensure a fair comparison.
>
> We aspire for our response to address the reviewer's concerns. We remain dedicated to addressing concerns you may possess with utmost eagerness.

---

> > ### Author Response · Authors · 2023-11-21
> >
> > Dear reviewer. I kindly inquire whether our response has successfully addressed your uncertainties. We remain dedicated to addressing concerns you may possess with utmost eagerness. Additionally, we would be encouraged if the reviewer considers raising the score.

---

> > > ### Comment · Reviewer_eFsa · 2023-11-21
> > > **Further Discussion**
> > >
> > > According to the experimental results, the proposed method is 'is clearly superior to state-of-the-art MD-based methods'. However, it same settings, FedAvg is actually performing best, such as $\omega = 1.0, 0.1$ on EMNIST, FMNIST and CIFAR10. Although the proposed method has higher privacy protection than FedAvg, it is difficult to measure the benefit. There are some methods using differential privacy mechanism to enhace privacy such as DPSGD [1],  Soteria[2] achieveing similar performance to FedAvg while preserving privacy. I would like to see the comparison between these methods.
> > >
> > > Reference:
> > >
> > > [1] Abadi M, Chu A, Goodfellow I, et al. Deep learning with differential privacy[C]//Proceedings of the 2016 ACM SIGSAC conference on computer and communications security. 2016: 308-318.
> > >
> > > [2] Sun J, Li A, Wang B, et al. Soteria: Provable defense against privacy leakage in federated learning from representation perspective[C]//Proceedings of the IEEE/CVF conference on computer vision and pattern recognition. 2021: 9311-9319.

---

### Official Review · Reviewer_iK9N · 2023-10-29

**Soundness:** 2 fair
**Presentation:** 3 good
**Contribution:** 2 fair
**Rating:** 5
**Confidence:** 3

**Summary:**

This work proposed a novel federated learning framework that aims to improve privacy. It achieves this by partitioning a classification network into two components: a feature extractor and a head. To obviate the need of synchronizing feature extractors across different clients, a conditional feature generator is introduced. The proposed framework offers an inherent advantage in privacy protection, as classical gradient inversion attacks become ineffective due to the absence of access to the feature extractor.

The conditional generator serves two purposes: it guides the training of local heads and also influences the output of local feature extractor. To further enhance local training and global aggregation, the authors incorporate various knowledge distillation technologies.

**Strengths:**

1. Federated learning is well-known for its privacy-preserving attributes but is also vulnerable to privacy attacks, such as gradient inversion attacks. In this context, exploring novel collaborative paradigms is crucial. This paper presents an intriguing approach to mitigating gradient inversion attacks by avoiding the synchronization of certain network component.

2. The proposed method incorporates a complex setup including 13 losses functions and 6 hyperparameters,  which appears overly complicated. However, the authors conduct an ablation study that justifies the inclusion of each individual loss. Additionally, they demonstrate that the method is relatively insensitive to the values of these hyperparameters.

**Weaknesses:**

1. Despite the second point mentioned in the Strengthens section, it appears that the proposed framework has a simpler substitute. For example, rather than training an additional feature generator, the clients can share their features and corresponding labels directly. In terms of privacy leakage, this alternative is equivalent to the proposed method, since the feature generator itself is trained on local features. Furthermore, direct feature sharing can largely reduce the framework's complexity.

2. Since the feature extractor is solely trained on the local device using the local data. The diversity of local data becomes critical for the training of feature extractor. Due to the same reason, I question the method's scalability to larger networks. As it stands, the paper only presents results based on the LeNet architecture.

3. The experimental results reveals that the proposed method underperforms compared to baseline approaches in the majority of evaluated settings.

4. Introducing an extra feature generator adds to the computational overhead. As acknowledged by the authors,  their method operates two to three times slower than baselines models.

**Questions:**

1. In Section 2.2, the authors claim "However, straightforward average aggregation may counteract the local knowledge from clients." Could you clarity this statement? Is it based on empirical evidence or theoretical reasoning?

2. Given that the method shows insensitivity to the hyperparameters $\lambda_1,\ldots,\lambda_6$ and that their valid value range includes 1, the author might consider removing these hyperparameters to simplify the proposed method.

---

> ### Author Response · Authors · 2023-11-14
>
> We highly appreciate Reviewer iK9N for positive comments and precious feedback on our work. We respond to specific comments below.
>
> Q1: Despite the second point mentioned in the Strengthens section, it appears that the proposed framework has a simpler substitute. For example, rather than training an additional feature generator, the clients can share their features and corresponding labels directly. In terms of privacy leakage, this alternative is equivalent to the proposed method, since the feature generator itself is trained on local features. Furthermore, direct feature sharing can largely reduce the framework's complexity.
>
> R1: During the preparation of the manuscript, we noted that there is existing work [1] that makes efforts on alternatives.
> This work focuses on each client extracts and sends to the server the means of local data representations and the corresponding soft predictions –- information that it refers to as “hyper-knowledge”.
> The server aggregates this information and broadcasts it to the clients in support of local training.
> It has previously been argued that communicating averaged data representation promotes privacy [2]; however, hyper-knowledge exchanged between server and clients may still be exposed to differential attacks [3, 4].
> Therefore, it promotes privacy with the help of the differential privacy mechanism.
>
> We argue that our approach is superior in terms of privacy leakage although it requires training an additional feature generator.
> Specifically, sharing the generator does not compromise privacy features of local data from clients, since the generator, which takes (z, y) as input, is intended to approximate the local feature extractor's outputs and avoid direct access to the original data x. Figure 4 in the paper verifies this statement. However, we do not deny the case where an attacker exploits the utility of the generator for inference attacks.
>
> Q2: Since the feature extractor is solely trained on the local device using the local data. The diversity of local data becomes critical for the training of feature extractor. Due to the same reason, I question the method's scalability to larger networks. As it stands, the paper only presents results based on the LeNet architecture.
>
> R2: We agree with this point about its application to intricate model scenarios. However, all our experiments are conducted on one NVIDIA Tesla V100 GPU with 32Gb of memory, thus running FedMD-CG and baselines on the intricate model would incur extreme time costs. Also, we follow existing work FedCG [2] and FedGen [3] to setup backbone architectures (i.e., LeNet-5) in our work. On this account, we concentrated on robust empirical validation, showcasing our method (FedMD-CG).  Our results, we believe, robustly demonstrate our method’s utility.
>
> Q3: The experimental results reveals that the proposed method underperforms compared to baseline approaches in the majority of evaluated settings.
>
> R3: We agree with the reviewer's comments. We acknowledge that our method underperforms compared to the baseline method FedAvg in the majority of evaluated settings.
> This is illustrated and discussed in the submitted paper. Please see Section 3.2 (Overview test performance comparison) in the main paper.
>
> In our work, we focus on model decoupling (MD)-based FL [5, 6, 7, 8], which has been widely acknowledged to mitigate the risk of privacy leakage of standard FL in the community. ln particular, our main contribution is to achieve better generalization performance given the same level of privacy protection as FedCG.  Additionally, our method significantly outperforms the MD-based baselines in most cases (see Table 1).
>
> Q4: In Section 2.2, the authors claim "However, straightforward average aggregation may counteract the local knowledge from clients." Could you clarity this statement? Is it based on empirical evidence or theoretical reasoning?
>
> R4: Thanks for the review's keen awareness. The premise scenario for this statement is the data heterogeneity scenario. Specifically, the local models are greatly drifted from each other in data heterogeneity scenario. Thus, traditional gradient (or model) averaging, i.e., straightforward average aggregation, could lose the knowledge in local models, and the performance of the updated global model is much lower than local models.
> The above statement is already a proven fact from existing works [9, 10, 11].
> We will revise and update this statement in the revised paper.

---

> > ### Author Response · Authors · 2023-11-14
> >
> > Q5: Given that the method shows insensitivity to the hyperparameters $\lambda_1,\ldots,\lambda_6$ and that their valid value range includes 1, the author might consider removing these hyperparameters to simplify the proposed method.
> >
> > R5: We agree with the reviewer's comments.
> > However, the insensitivity of our method to the hyperparameters $\lambda_1,\ldots,\lambda_6$ is verified by conducting experiments.
> > During the preparation of the manuscript, we were unable to specify hyperparameters $\lambda_1,\ldots,\lambda_6$ that made our method optimal.
> > So, the hyperparameters $\lambda_1,\ldots,\lambda_6$ are necessary for the description of our method.
> >
> > We are honored to address your uncertainties and concerns. We remain dedicated to addressing concerns you may possess with utmost eagerness.
> >
> > [1] Chen H, Vikalo H. The Best of Both Worlds: Accurate Global and Personalized Models through Federated Learning with Data-Free Hyper-Knowledge Distillation[J]. arXiv preprint arXiv:2301.08968, 2023.
> >
> > [2] Yue Tan, Guodong Long, Lu Liu, Tianyi Zhou, Qinghua Lu, Jing Jiang, and Chengqi Zhang. Fedproto: Federated prototype learning over heterogeneous devices. arXiv preprint arXiv:2105.00243, 2021.
> >
> > [3] Cynthia Dwork. Differential privacy: A survey of results. In International conference on theory and applications of models of computation, pp. 1–19. Springer, 2008.
> >
> > [4] Robin C Geyer, Tassilo Klein, and Moin Nabi. Differentially private federated learning: A client level perspective. arXiv preprint arXiv:1712.07557, 2017.
> >
> > [5] Wu, Y., Kang, Y., Luo, J., He, Y., & Yang, Q. (2021) Fedcg: Leverage conditional gan for protecting privacy and maintaining competitive performance in federated learning. arXiv preprint arXiv:2111.08211.
> >
> > [6] Zhu, Z., Hong, J., & Zhou, J. (2021) Data-free knowledge distillation for heterogeneous federated learning. In International Conference on Machine Learning, pp. 12878-12889.
> >
> > [7] Arivazhagan, M. G., Aggarwal, V., Singh, A. K., & Choudhary, S. (2019) Federated learning with personalization layers. arXiv preprint arXiv:1912.00818.
> >
> > [8] Liang, P. P., Liu, T., Ziyin, L., Allen, N. B., Auerbach, R. P., Brent, D., Salakhutdinov R. & Morency, L. P. (2020) Think locally, act globally: Federated learning with local and global representations. arXiv preprint arXiv:2001.01523.
> >
> > [9] Lin Zhang, Li Shen, Liang Ding, Dacheng Tao, and Ling-Yu Duan. Fine-tuning global model via data-free knowledge distillation for non-iid federated learning. In CVPR, pp. 10174–10183, 2022c.
> >
> > [10] Sai Praneeth Karimireddy, Satyen Kale, Mehryar Mohri, Sashank Reddi, Sebastian Stich, and Ananda Theertha Suresh. Scaffold: Stochastic controlled averaging for federated learning. In International Conference on Machine Learning, pp. 5132–5143. PMLR, 2020.
> >
> > [11] Kangyang Luo, Xiang Li, Yunshi Lan, and Ming Gao. Gradma: A gradient-memory-based accelerated federated learning with alleviated catastrophic forgetting. In Proceedings of the IEEE/CVF Conference on Computer Vision and Pattern Recognition, pp. 3708–3717, 2023.

---

> > > ### Author Response · Authors · 2023-11-21
> > >
> > > Dear reviewer. I kindly inquire whether our response has successfully addressed your uncertainties. We remain dedicated to addressing concerns you may possess with utmost eagerness. Additionally, we would be encouraged if the reviewer considers raising the score.

---

### Official Review · Reviewer_i6FV · 2023-11-04

**Soundness:** 3 good
**Presentation:** 3 good
**Contribution:** 2 fair
**Rating:** 5
**Confidence:** 5

**Summary:**

The paper proposes a federated learning algorithm that primarily attempts to improve privacy preservation (mitigate deep gradient leakage attacks by a semi-honest server). The main idea is to decompose the model into feature extractor and classifier components. While the classifier component is aggregated using the standard FedAvg approach, the local feature extractors are replaced by local conditional generators, which are then aggregated by the server into a global generator. In each collaborative round, the clients alternatively update their feature extractors (through knowledge distillation from the global generator) and local generators (through knowledge distillation from the local feature extractor and classifier).

**Strengths:**

1) The proposed idea closely follows the FedCG framework (Wu et al. 2021), but proposes several incremental innovations to improve upon the FedCG framework. Though the proposed changes appear to be complicated, they are simply based on knowledge distillation concepts and most of them appear to be intuitively reasonable.

2) The paper is well-written and self-contained. The presentation style is also easy to follow.

**Weaknesses:**

1) The foremost goal of the paper is to mitigate privacy leakage through deep gradient leakage (DLG) attacks by a semi-honest server. As noted in the appendix, it is hard to come up with theoretical privacy guarantees for this framework. However, it is certainly feasible to come up with strong attacks and evaluate the proposed algorithm against such strong attacks. While some leakage results are reported in Table 2 and Figure 4, the paper fails to specify the exact attack mechanism deployed. In the proposed framework, the only unknown (compared to the FedAvg setting) for a malicious server are the parameters of the local feature extractors in each round. It should be possible to come up with sophisticated DLG attacks following the same knowledge distillation strategies employed for client training. For instance, can a reverse distillation process be used by the server to transfer knowledge from the local generators to estimate the local feature extractors and then use these local feature extractors to reconstruct the input samples? Can the above attack benefit from the availability of an auxiliary dataset at the server?

2) The evaluation metrics (local and global accuracy) used in this work appear to be unrealistic. The local partition of the test is stochastic and not reproducible. Given that no statistics has been reported, it is hard to judge the variations in performance. On the other hand, the global accuracy also does not make sense because that requires constructing a "virtual" model using one round of FedAvg, which is against the main principle of the proposed method. Maybe the local accuracy at each client could be computed based on all the test data and average of these local accuracy values should be reported along with standard deviation.

3) The model used for the experiments is LeNet-5, which is not a reasonable model for most real-world applications. Furthermore, this model has been arbitrarily divided into feature extractor and classifier. The main argument of this work is that the latent feature space has so-called "high-level patterns" and it is difficult to construct the input data from these patterns. How accurate is this assumption when the feature extractor is simply a couple of conv layers? Moreover, the datasets used in this experiment are also not representative (very low resolutions images with utmost 10 classes). How will the proposed approach scale for larger resolution images (e.g., 224 x 224) with deeper neural networks (e.g., ResNet-18) and more classes (e.g., ImageNet)?

**Questions:**

Please see weaknesses.

---

> ### Author Response · Authors · 2023-11-14
>
> We highly appreciate Reviewer i6FV for positive comments and precious feedback on our work. We respond to specific comments below.
>
> Q1:  For instance, can a reverse distillation process be used by the server to transfer knowledge from the local generators to estimate the local feature extractors and then use these local feature extractors to reconstruct the input samples? Can the above attack benefit from the availability of an auxiliary dataset at the server?
>
> R1: We argue that sharing the generator instead of the feature extractor itself does not compromise privacy features of local data from clients, since the generator, which takes (z, y) as input, is intended to approximate the local feature extractor's outputs and avoid direct access to the original data x. Figure 4 in the paper verifies this statement. However, we do not deny the case where an attacker exploits the utility of the generator for inference attacks.
> Moreover, in our empirical experiments, we actually used model inversion to attack the generator.
>
>
> Q2: The evaluation metrics (local and global accuracy) used in this work appear to be unrealistic. The local partition of the test is stochastic and not reproducible. Given that no statistics has been reported, it is hard to judge the variations in performance. On the other hand, the global accuracy also does not make sense because that requires constructing a "virtual" model using one round of FedAvg, which is against the main principle of the proposed method. Maybe the local accuracy at each client could be computed based on all the test data and average of these local accuracy values should be reported along with standard deviation.
>
> R2: The L.acc values reported in our paper are similar to the review's suggested strategy, which are to compute local accuracies for each client based on the test data that has been divided equally and to report the mean and standard deviation of these local accuracies.
> In other words, the distribution of the test data set divided into each client is consistent with that of all test data.
> In addition, we agree with the review's comments on G.acc. It is true that G.acc cannot be computed in real-world scenarios, but at the same time the local model of each client does not have access to all test data.
> In our experiments, we construct "virtual" models only to assist us in comprehensively evaluating the superiority of our method.
>
> Moreover, the empirical results in our paper can all be reproduced by the code we provide.
>
> Q3: The model used for the experiments is LeNet-5, which is not a reasonable model for most real-world applications. Furthermore, this model has been arbitrarily divided into feature extractor and classifier. The main argument of this work is that the latent feature space has so-called "high-level patterns" and it is difficult to construct the input data from these patterns. How accurate is this assumption when the feature extractor is simply a couple of conv layers? Moreover, the datasets used in this experiment are also not representative (very low resolutions images with utmost 10 classes). How will the proposed approach scale for larger resolution images (e.g., 224 x 224) with deeper neural networks (e.g., ResNet-18) and more classes (e.g., ImageNet)?
>
> R3: We agree with this point about its application to intricate data and model scenarios. However, all our experiments are conducted on one NVIDIA Tesla V100 GPU with 32Gb of memory, thus running FedMD-CG and baselines on the intricate data and model would incur extreme time costs. Also, we follow existing work FedCG [1] and FedGen [2] to setup backbone architectures (i.e., LeNet-5) in our work. On this account, we concentrated on robust empirical validation, showcasing our method (FedMD-CG).
>
> We are honored to address your uncertainties and concerns. We remain dedicated to addressing concerns you may possess with utmost eagerness.
>
> [1] Wu, Y., Kang, Y., Luo, J., He, Y., & Yang, Q. (2021) Fedcg: Leverage conditional gan for protecting privacy and maintaining competitive performance in federated learning. arXiv preprint arXiv:2111.08211.
>
> [2] Zhu, Z., Hong, J., & Zhou, J. (2021) Data-free knowledge distillation for heterogeneous federated learning. In International Conference on Machine Learning, pp. 12878-12889.

---

> > ### Author Response · Authors · 2023-11-21
> >
> > Dear reviewer. I kindly inquire whether our response has successfully addressed your uncertainties. We remain dedicated to addressing concerns you may possess with utmost eagerness. Additionally, we would be encouraged if the reviewer considers raising the score.

---

### Official Review · Reviewer_C5Ch · 2023-11-05

**Soundness:** 3 good
**Presentation:** 3 good
**Contribution:** 3 good
**Rating:** 6
**Confidence:** 3

**Summary:**

This paper proposes that FedMD-CG improves the privacy protection capability of FL. This work uses a condition generator instead of a feature extractor to perform server-side model aggregation. In this work, KD is used to ensure the consistency of local generator and classifier. The server uses KD to aggregate trained local generators and classifiers. This paper has done a lot of empirical experimental research and discussion.

**Strengths:**

This paper introduces in detail how to design loss and training methods on client and server side, and gives a comprehensive experience experiment. The proposed scheme makes trade-offs in terms of performance and privacy protection.

**Weaknesses:**

There are some unclear descriptions in the paper that are confusing. For example, the form of the L2-norm function in formulas (2) and (7) is inconsistent.

**Questions:**

At the end of page 3 of the paper, the author claims to design two loss functions. to transfer the knowledge of the local model to the local generator. Does this imply that the losses in formulas (5) and (6) will only find gradients for generators?

When the server trains its own model, does it need the label y to calculate losses and gradients? Does getting the distribution of labels reveal privacy?

---

> ### Author Response · Authors · 2023-11-14
>
> We highly appreciate Reviewer C5Ch for positive comments and precious feedback on our work. We respond to specific comments below.
>
> Q1: There are some unclear descriptions in the paper that are confusing. For example, the form of the L2-norm function in formulas (2) and (7) is inconsistent.
>
> R1: Thanks for the review's keen awareness.  They are consistent, except that the L2-norm in Eq. (2) needs to be squared, while the L2-norm in Eq. (7) does not. For ease of understanding, we will add the subscript 2 to Eq. (2) and update it in the revised paper.
>
> Q2: At the end of page 3 of the paper, the author claims to design two loss functions. to transfer the knowledge of the local model to the local generator. Does this imply that the losses in formulas (5) and (6) will only find gradients for generators?
>
> R2: No, it dose not. During the experiments, gradients of generators and classifiers can be obtained by backpropagation, but we only update generators at this stage.
>
> Q3: When the server trains its own model, does it need the label y to calculate losses and gradients? Does getting the distribution of labels reveal privacy?
>
> R3: Label y is needed to compute loss and gradient when the server trains its own model. Whether getting the distribution of labels compromises privacy is discussed in Appendix F, please see Appendix F.
>
> We are honored to address your uncertainties and concerns. We remain dedicated to addressing concerns you may possess with utmost eagerness.

---

### Author Response · Authors · 2023-11-14

We thank all reviewers for their thoughtful, constructive and positive review of our manuscript. We are encouraged to hear that the reviewers found the FedDM-CG method we present to be interesting and practical (Reviewers i6FV, iK9N, eFsa), and thoroughly-evaluated (Reviewers C5Ch, iK9N). Meanwhile, they view our manuscript as well-written (Reviewer i6FV). In response to the feedback, we provide detailed responses to address each reviewer's concerns below.